# Reduced ADP off-rate by the yeast CCT2 double mutation T394P/R510H which causes Leber congenital amaurosis in humans

Mousam Roy[1,2], Rachel C. Fleisher [1,2], Alexander I. Alexandrov[1] & Amnon Horovitz [1✉]

The CCT/TRiC chaperonin is found in the cytosol of all eukaryotic cells and assists protein folding in an ATP-dependent manner. The heterozygous double mutation T400P and R516H in subunit CCT2 is known to cause Leber congenital amaurosis (LCA), a hereditary congenital retinopathy. This double mutation also renders the function of subunit CCT2, when it is outside of the CCT/TRiC complex, to be defective in promoting autophagy. Here, we show using steady-state and transient kinetic analysis that the corresponding double mutation in subunit CCT2 from *Saccharomyces cerevisiae* reduces the off-rate of ADP during ATP hydrolysis by CCT/TRiC. We also report that the ATPase activity of CCT/TRiC is stimulated by a non-folded substrate. Our results suggest that the closed state of CCT/TRiC is stabilized by the double mutation owing to the slower off-rate of ADP, thereby impeding the exit of CCT2 from the complex that is required for its function in autophagy.

[1] Department of Chemical and Structural Biology, Weizmann Institute of Science, Rehovot 7610001, Israel. [2] These authors contributed equally: Mousam Roy, Rachel C. Fleisher. ✉email: Amnon.Horovitz@weizmann.ac.il

Leber congenital amaurosis (LCA) is an inherited retinal dystrophy, which is responsible for about 20% of cases of childhood blindness[1,2]. LCA is currently associated with mutations in 38 genes. Some of the most common LCA-linked mutations are in genes that are expressed solely or predominantly in the retina or the retinal pigment epithelium such as those coding for guanylate cyclase-1, retinoid isomerase, inosine 5'-monophosphate dehydrogenase 1 and retinol dehydrogenase 12[1,2]. Interestingly, however, it was reported that the heterozygous double mutation T400P and R516H in subunit CCT2 of the ubiquitously expressed CCT/TRiC chaperonin is also LCA-causative[3]. The CCT/TRiC chaperonin, which is found in the cytosol of all eukaryotic cells, assists protein folding in an ATP-dependent manner[4-8]. It is comprised of two back-to-back stacked oligomeric rings each with a cavity in which protein folding can take place under protective conditions[4-8]. Each ring of CCT/TRiC is made up of eight distinct subunits, CCT1 to CCT8, which are arranged in a fixed order around the ring. CCT/TRiC cycles between an open apo state, which can bind clients via subunit-specific contacts, and a closed state reached following cooperative ATP binding and hydrolysis in which the client becomes encapsulated[4-8].

The folding function of CCT/TRiC is believed to require the intact oligomeric complex but individual subunits may also have moonlighting roles as suggested in early studies[9,10] by the discovery of some of them in the nucleus. Recently, for example, it was reported that monomeric subunit CCT2 interacts with autophagosome marker ATG8 family members, thereby promoting the autophagic degradation of solid protein aggregates[11]. This interaction was found to be mediated by the human CCT2 residues VLL at positions 503-505 and VIL at positions 513-515, which become exposed upon dissociation of subunit CCT2 from the rest of the CCT/TRiC complex. The proximity of the site of the LCA-associated mutation R516H to these ATG8-interacting residues suggested that it may cause disease by interfering with autophagic membrane targeting and solid aggregate degradation. A link between LCA and impaired autophagy has indeed been reported before[12]. Both mutations, R516H and T400P, were found to reduce the association of CCT2 with Atg8[11] but the mechanism by which this is caused by the mutation T400P remained less clear. Given that the mutation T400P is in helix 14, which contains a conserved catalytic aspartic residue, it was speculated[11] that T400P alters CCT/TRiC's ATPase activity and stabilizes its closed state. Stabilization of the closed state would impede subunit CCT2 dissociation from the complex, thereby impairing autophagy. Here, we tested this hypothesis by carrying out a detailed analysis of the ATPase activity of CCT/TRiC from *Saccharomyces cerevisiae* with the mutations T394P and R510H in subunit CCT2.

## Results and discussion

Residues T394 and R510 in *S. cerevisiae* CCT2, which correspond to residues T400 and R516 in human CCT2, are located in helices 14 and 18, respectively (Fig. 1a). These helices, like the rest of CCT2, are highly conserved in human, yeast and other organisms (Fig. 1b). Comparing the cryo-EM structure of human CCT2 and the crystal structure of yeast CCT2, both in a nucleotide-bound state, shows that they are very similar (Fig. 1c), as expected given the high sequence identity, and suggests that effects of mutations in yeast CCT2 on ATPase activity may mimic those in human CCT2. Moreover, these mutations were found to interfere with autophagy in yeast (Supplementary Fig. 1) as reported before for humans[11]. Taken together, these data indicate that analyzing the effects of these mutations in yeast CCT/TRiC can provide insight into their effects in humans.

The steady-state ATPase activities of wild-type CCT/TRiC, the T394P and R510H single mutants, and the corresponding double mutant were characterized by measuring initial rates of ATP hydrolysis as a function of ATP concentration (Fig. 2). The curves for the wild-type and the two single mutants, which were found to have similar profiles, were fitted to Eq. 1 for two allosteric transitions. In the case of the wild-type, the values of the apparent ATP binding constants, $K_1$ and $K_2$, were found to be 8.3 ($\pm$1.6) and 110 ($\pm$39) $\mu$M, respectively, and the values of the Hill constants for the first and second transitions, $n$ and $m$, were found to be 1.5 ($\pm$0.3) and 3.0 ($\pm$1.9), respectively. These values are in excellent agreement with the respective values of 9.6 ($\pm$0.4) and 97 ($\pm$58) $\mu$M for $K_1$ and $K_2$ and the respective values of 1.2 ($\pm$0.6) and 3.0 ($\pm$0.2) for $n$ and $m$, which were reported before[13]. The values of $k_{cat}$ for ATP hydrolysis by one and both rings were found to be 0.043 ($\pm$0.005) and 0.053 ($\pm$0.001) s$^{-1}$, respectively, and, thus, close to those reported before[13]. In the case of the R510H single mutant, the values of the apparent ATP binding constants, $K_1$ and $K_2$, were found to be 7.1 ($\pm$2.4) and 31 ($\pm$4) $\mu$M, respectively, and the values of the Hill constants for the first and second transitions, $n$ and $m$, were found to be 2.1 ($\pm$0.9) and 5.3 ($\pm$4), respectively. The values of $k_{cat}$ for ATP hydrolysis by one and both rings of this mutant were found to be 0.10 ($\pm$0.03) and 0.150 ($\pm$0.002) s$^{-1}$, respectively. In the case of the T394P single mutant, the values of the apparent ATP binding constants, $K_1$ and $K_2$, were found to be 8.9 ($\pm$3.4) and 66 ($\pm$24) $\mu$M, respectively, and the values of the Hill constants, $n$ and $m$, were found to be 1.7 ($\pm$0.6) and 3.2 ($\pm$2.0), respectively. The values of $k_{cat}$ for ATP hydrolysis by one and both rings of this mutant were found to be 0.11 ($\pm$0.03) and 0.150 ($\pm$0.003) s$^{-1}$, respectively.

Strikingly, however, the curve for the double mutant has a small but reproducible peak at about 150 $\mu$M ATP that is absent in the data for the single mutants (Fig. 2b) and has not been reported before for wild-type CCT/TRiC. The curve for the double mutant is, however, qualitatively similar to the biphasic curves of wild-type GroEL[14]. In the case of GroEL, the decrease in ATPase activity at higher ATP concentrations has been attributed to the slow off-rate of ADP and was accordingly found to be absent in the E257A GroEL mutant with a fast off-rate for ADP[15]. The distinct curve of the double mutant compared to those of the single mutants is in agreement with the study by Minegishi et al. [3] that indicated that heterozygous individuals with both mutations suffer from LCA whereas individuals with one wild-type allele and one mutant allele (T400P or R516H) do not (although the double mutant in our work contains both mutations in both rings (Supplementary Fig. 2)). Given the link suggested by these results between LCA caused by the double mutation and a slow off-rate for ADP, we decided to further test whether the off-rate for ADP is indeed slower in the double mutant.

The ATPase activities of wild-type CCT/TRiC and the double mutant were examined in more detail by carrying out transient kinetic experiments. Equal volumes of CCT/TRiC and different concentrations of ATP were rapidly mixed using a stopped-flow device, and the time-resolved changes in inorganic phosphate concentration upon ATP hydrolysis were followed by measuring the change in fluorescence of a coumarin-labeled phosphate-binding protein (PBP)[16]. For each ATP concentration, 5–10 traces (each with 2000 data points) were averaged and fitted to Eq. S14 in Supplementary Note 1. The average traces of wild-type CCT/TRiC and the double mutant all consist, as observed before[17], of two burst phases followed by a lag phase and then a steady-state phase, which are described by three respective exponential terms and a linear term in Eq. S14.

Plots of the rates and amplitudes of the two burst phases for wild-type CCT/TRiC and the double mutant are shown in Fig. 3. Three lines of evidence indicate that the off-rate of ADP of the

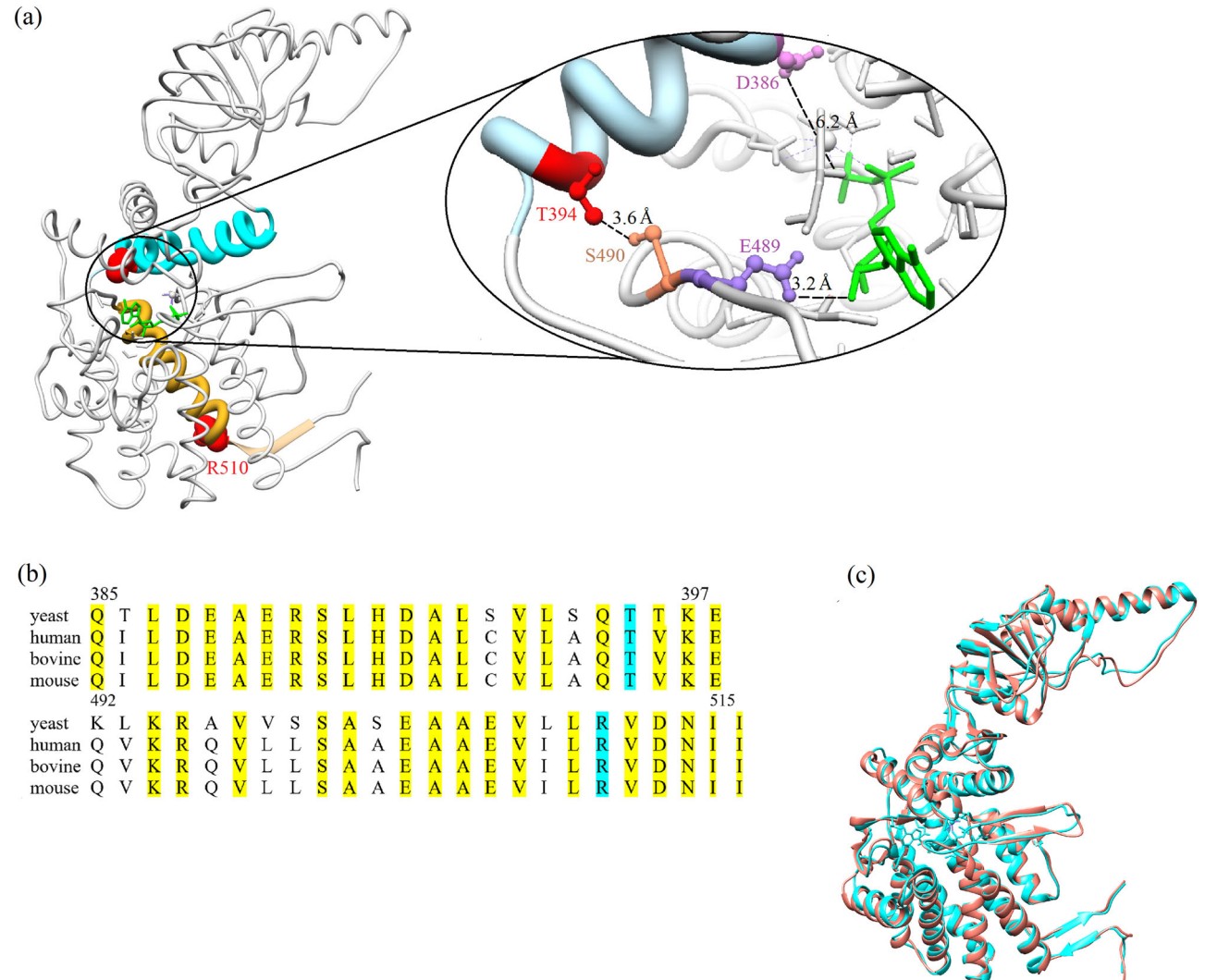

**Fig. 1 Structure and sequence comparisons of subunit CCT2 in CCT/TRiC from different organisms. a** Crystal structure of yeast subunit CCT2 (PDB ID: 4V8R) in the ADP-bound state showing the mutated residues (red), helices 14 (cyan) and 18 (gold), and ADP (green). The contact site between helix 14 and ADP is magnified. It can be seen that a mutation at position 394 can have an allosteric effect on E489, which is in contact with ADP, via S490. A mutation at position 510 can affect ADP via a rigid-body movement of helix 18. Also shown is D386 (pink), which is a conserved catalytic residue. **b** Alignment of the sequences of helix 14 (top) and helix 18 and strand 25 (bottom) in subunit CCT2 from different organisms. The mutated positions are highlighted in cyan and other conserved positions in yellow. The numbering corresponds to yeast CCT2. The overall amino acid sequence identities of yeast and human helix 14, helix 18 and strand 25 and the entire CCT2 subunit are about 78, 71, and 65%, respectively. The corresponding sequence similarities are about 87, 92 and 80%, respectively. **c** Superimposed cryo-EM structure of human CCT2 (PDB ID: 7LUM) and the crystal structure of yeast CCT2 (PDB ID: 4V8R) both in the closed state. The structures are very similar as indicated by a root mean square deviation of ~1.2 Å. Panels a and c were generated using Chimera version 1.15. The single-letter code of amino acids is used.

double mutant is slower than that of the wild-type. First, it can be seen that the amplitudes of the burst phases of the double mutant are larger than those of the wild-type (Fig. 3b). In the case of fast ADP release, the initial round of ATP hydrolysis, which does not involve prior ADP release, will not differ much from subsequent rounds that require ADP release, thereby resulting in burst phases with diminished amplitudes. Second, it may be seen that the rates of the burst phases of the double mutant increase in a hyperbolic fashion with increasing ATP concentrations whereas those of the wild-type decrease (Fig. 3a). These increasing and decreasing dependencies on ligand (ATP) concentrations are consistent with simple binding (or induced fit) and conformational selection mechanisms of ligand binding, respectively[18], as reported before[17] for wild-type CCT/TRiC and shown for the double mutant in Fig. 4. The apparent ligand affinity in the case of simple binding (or induced fit) is higher than for conformational

selection[18], thereby suggesting that the off-rate of ADP of the double mutant is lower than that of the wild-type. Third, the fit of the data in Fig. 3a for the double mutant and wild-type to expressions for the observed rate constants (see ref. [18]. and legend to Fig. 3) in cases of simple binding and conformational selection, respectively, provide estimates of the ligand affinities. In the case of the double mutant, the ATP affinities corresponding to the two burst phases are similar and found to be about 7 μM whereas in the case of the wild-type they are lower (about 27 and 250 μM), thereby indicating again that the off-rate of ADP of the double mutant is lower than that of the wild-type. Here, we have assumed that an increase in affinity for ATP indicates an increase also in the affinity of ADP as commonly observed, for example, in response to potassium concentration[19].

The ATPase activity of GroEL is stimulated by non-folded substrates, such as reduced and calcium-depleted α-lactalbumin[20],

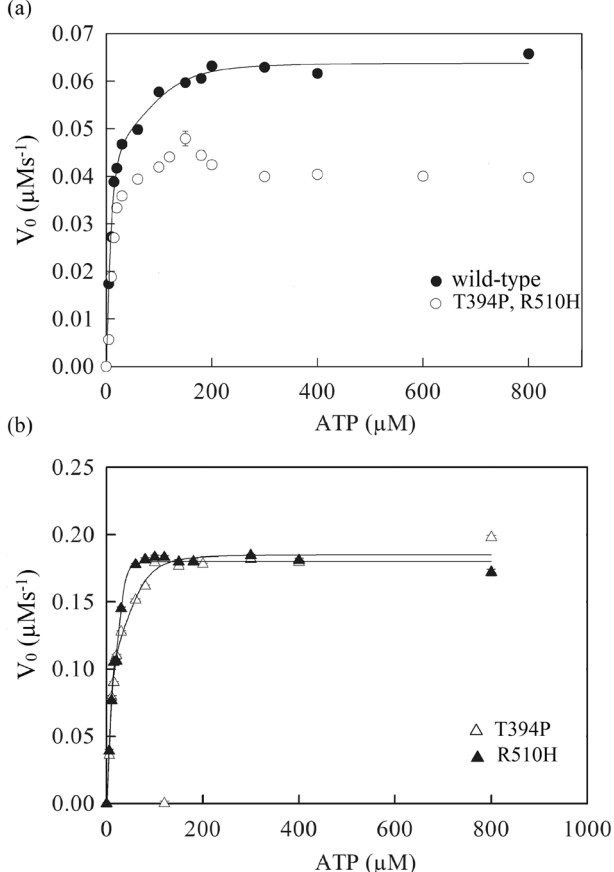

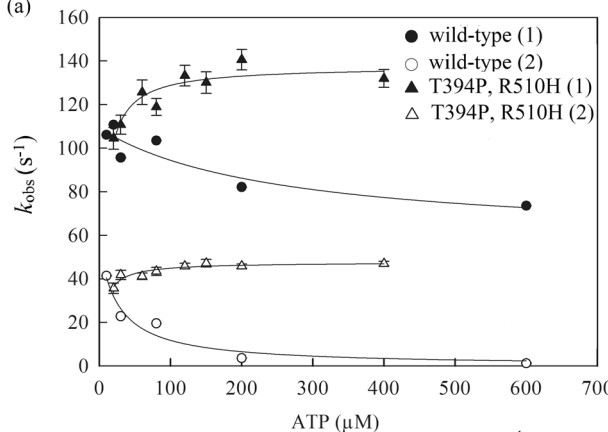

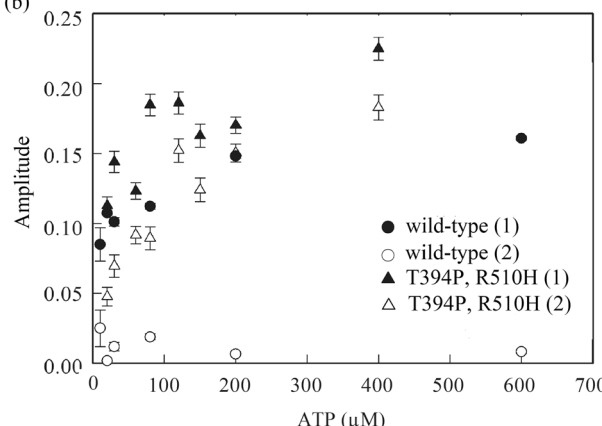

**Fig. 2 Steady-state ATPase activity of wild-type CCT/TRiC, the single mutants, T394P and R510H, in subunit CCT2 and the corresponding double mutant at different concentrations of ATP.** Initial velocities of ATP hydrolysis by wild-type CCT/TRiC and the different mutants were measured at different concentrations of ATP using a final oligomer concentration of CCT/TRiC of 20 nM. The data for wild-type CCT/TRiC (**a**) and the single mutants (**b**) were fitted to Eq. 1 for two allosteric transitions. See Materials and Methods for more details. Error bars represent standard errors.

**Fig. 3 Plots of the values of the observed rate constants and amplitudes of the two burst phases of wild-type CCT/TRiC and the double mutant as a function of ATP concentration.** Values of the observed rate constants (**a**) and amplitudes (**b**) for the two burst phases (designated 1 and 2) were obtained by fitting the averaged traces to Eq. S14. The data in (**a**) for each of the observed rate constants of the double mutant were fitted, in accordance with a simple binding mechanism, to $k_{obs} = k_{cat}[ATP]^m/([ATP]^m + K_d)$, where $m$ and $K_d$ are the Hill and ATP dissociation constants. The data in (**a**) for each of the observed rate constants of the wild-type were fitted as before[17], in accordance with the conformational selection mechanism, to $k_{obs} = k_{-1}K_d/([ATP]^m + K_d) + k_1$, where $k_1$ and $k_{-1}$ are the forward and reverse rate constants for the conformational change. Error bars represent standard errors.

but a similar effect has not been reported for CCT/TRiC in the case of any substrate. Given that the stimulation of GroEL's ATPase activity by substrates appears to be due to an increase in the ADP off-rate[15], we reasoned that the ATPase activity of the CCT/TRiC double mutant might be stimulated more than that of wild-type since it has a lower ADP off-rate. The steady-state ATPase activities of the wild-type CCT/TRiC and double mutant were, therefore, measured as a function of α-lactalbumin concentration at a fixed ATP concentration of 300 µM (Fig. 5). The results show that the ATPase activity of CCT/TRiC is stimulated by a non-folded protein substrate as in the case of GroEL[20] and that this effect is indeed greater in the case of the double mutant, thus providing further evidence that its ADP off-rate is slower. Dynamic light scattering experiments show that α-lactalbumin binding to wild-type CCT/TRiC and the double mutant does not lead to their destabilization (Supplementary Fig. 3).

In summary, the steady-state and transient kinetic data reported here indicate that the off-rate of ADP of CCT/TRiC containing T394P/R510H in subunit CCT2 (which corresponds to the LCA-causative double mutation in humans) is slower than that of wild-type CCT/TRiC, thereby leading to stabilization of the closed state of the CCT/TRiC double mutant. The buried surface area of CCT2 in the CCT/TRiC closed state (PDB ID:

4V8R) is calculated[21] to be about 3600 Å² more than in the open state (PDB ID: 5GW4), which corresponds to about 6 kcal mol⁻¹ more binding energy[22] in the closed vs. open states. Contact maps show that CCT2 also has many more inter-subunit interactions in the closed vs. open states (Supplementary Fig. 4). A more long-lived closed state owing to a slower ADP off-rate would, thus, impede the exit of subunit CCT2 from the complex and impair its function outside the complex in autophagy[11]. A two-fold slower ADP off-rate, as indicated by the effect of α-lactalbumin on ATPase activity (Fig. 5), can be estimated to reduce the amount of free CCT2 by a factor of two (Eq. S15 in Supplementary Note 1). It is important to note, however, that the closed state is reached post-ATP hydrolysis (as in our experiments) and not upon adding ADP.

It was not clear from the previous work[3] whether heterozygotes suffer from LCA because all of their CCT2 subunits contain one of the two mutations or because a proportion (e.g. 50%) of their CCT/TRiC complexes contain both mutations (Supplementary

$$T_a + ATP \underset{k_{-1}}{\overset{k_1}{\rightleftharpoons}} T_aATP_m \overset{P_i}{\underset{k_{cat1}}{\longrightarrow}} T_aADP_m$$

$$R + ATP \underset{k_{-5}}{\overset{k_5}{\rightleftharpoons}} RATP_q \overset{}{\underset{k_{cat3}}{\longrightarrow}} R + Pi$$

$$T_b + ATP \underset{k_{-2}}{\overset{k_2}{\rightleftharpoons}} T_bATP_n \overset{P_i}{\underset{k_{cat2}}{\longrightarrow}} T_bADP_n$$

**Fig. 4 Scheme showing various allosteric states of the CCT/TRiC double mutant. In this scheme, T and R stand for the apo states of CCT/TRiC with low and high affinities for ATP, respectively, and P designates the inorganic phosphate product.** $T_a$ and $T_b$ designate different subsets of subunits in the T state. For simplicity, it is assumed that $T_a$, $T_b$, and R bind m, n, and q molecules of ATP, respectively, which then undergo hydrolysis. The two burst phases reflect ATP hydrolysis by the $T_a$ and $T_b$ states, respectively.

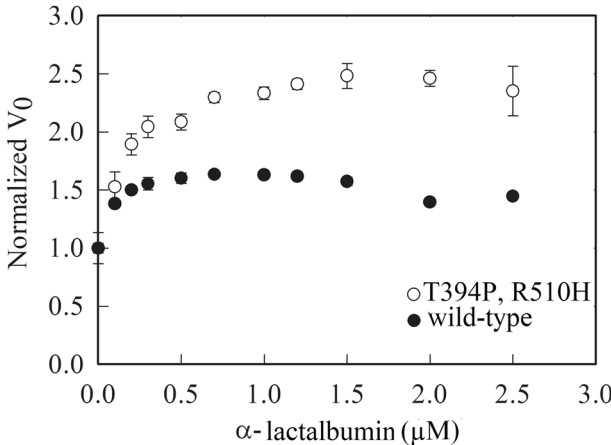

**Fig. 5 Stimulation of the steady-state ATPase activity of wild-type CCT/TRiC and the double mutant by α-lactalbumin.** Initial velocities of ATP hydrolysis by the tagged wild-type CCT/TRiC and the double mutant were measured as a function of α-lactalbumin concentration at a fixed ATP concentration of 300 μM. The final oligomer concentration of CCT/TRiC was 20 nM. The data for each variant were normalized by its steady-state ATPase activity in the absence of α-lactalbumin. See Materials and Methods for more details. Error bars represent standard errors.

Fig. 2). Given that the single mutants in our study don't show altered ADP off-rates and that only heterozygotes suffer from LCA, our results indicate that the altered off-rate in the double mutant suggested being LCA-causative is due to inter-ring communication between the different mutations. Such communication is likely to be facilitated by the direct interaction of subunit CCT2 in the two rings but the mechanism underlying it still needs to be established. LCA caused by this double mutation in CCT2 may, therefore, be an example of both a congenital disorder of autophagy[23] and an allosteric disease. In zebrafish, a different perturbation of helix 14 in CCT2 (L394H and deletion of residues at positions 395-401) was also found to affect eye development[24]. It remains to be established whether the effect in zebrafish, which was attributed to cell cycle defects and cell death, also involves impaired autophagy and inter-ring allostery.

## Methods

**Molecular biology**. Mutagenesis was carried out by RF cloning[25] using the pET17b plasmid harboring the gene coding for *S. cerevisiae* subunit CCT2 followed downstream by the kanamycin resistance cassette and the gene's 3′ untranslated region (UTR). The following mutagenic forward primers were used:

T394P: 5'-CGTGTTGTCACAGCCAACAAAGGAAACAAG-3'
R510H: 5'-GCTGAAGTTCTACTACATGTGGATAACATCATCC-3'

together with the respective reverse primers: 5'-GTAGAACTTCAGCGGCTTC-3' and 5'-GGATGTGATGTGAGAACTGTATCC-3' to generate megaprimers for amplifying the entire plasmids with the desired mutations. The megaprimer for creating the plasmid with the double mutant was generated using the mutagenic forward primer for T394P and reverse primer for R510H. The plasmids were then digested with XbaI and NotI-HF to generate linear fragments that contain the CCT2 gene with the double or one of the single mutations. The mutations were introduced into the chromosomal gene for subunit CCT2 via homologous recombination by transforming haploid *S. cerevisiae* yeast cells (strain BJ2168) with these fragments. In these cells, the chromosomal *CCT6* gene was replaced with a copy containing an in-frame insertion of the coding sequence for calmodulin binding peptide (CBP), followed by a selectable marker (URA3) downstream of the gene[26]. The genotype of this strain is *Mat a leu2, trp1, gal2, ura3-52, leu2-3, prb1-1122, pep4-3, prc467, CCT6::CBP::URA3*. Colonies containing the desired mutations were identified by sequencing the entire gene for CCT2. The mutations were found to have relatively small effects on cell growth under normal conditions (Supplementary Fig. 5).

**CCT/TRiC expression and purification**. Cells containing wild-type CCT/TRiC or the mutants were grown in 10 L YPD medium containing 300 μg/mL geneticin at 30 °C. CCT/TRiC was then purified essentially as described earlier[17]. In brief, cell pellets were resuspended in 300 mM Hepes buffer (pH 8.0) containing 200 mM NaCl, 35% glycerol, 2 mM EDTA and 4 mM DTT. The cells were then lysed in the presence of 0.01 mg/mL chymostatin, 0.01 mg/mL antipain, 0.01 mg/mL pepstatin A, 0.02 mg/mL leupeptin and a protease inhibitor cocktail (Apex Bio, K1007) diluted 1:500. The cell lysate was spun at 17,600 g for 30 min. The supernatant was then passed through a Whatman filter paper and supplemented with 5 mM CaCl₂, 0.01% LDAO, and aprotinin solution (Sigma A6279) at a 1:500 dilution and an additional 2 mM DTT. The filtrate was then incubated overnight at 4 °C with calmodulin beads (GE Healthcare Calmodulin Sepharose 4B) that had been equilibrated with 20 mM Hepes buffer (pH 8.0) containing 1 M NaCl, 2 mM CaCl₂, 2 mM DTT, 0.01% LDAO and 20% glycerol (EQ buffer). The beads were then packed in a column and the flow-through was discarded. The column was then washed with 70 mL EQ buffer, 70 mL wash buffer (identical to EQ buffer but with 0.2 mM CaCl₂ and 200 mM NaCl) containing 5 mM ATP and 20 mM MgCl₂, and then 130 mL wash

buffer (without ATP and MgCl$_2$) for removing the ATP. Care was taken that the flow rate did not exceed 0.2 mL/min during the washing. Elution was carried out with a wash buffer containing 10 mM EDTA (without CaCl$_2$). All the fractions containing the protein were loaded on a sucrose gradient and centrifuged at 112,700 g for 64–68 h. The fractions from the gradient that contains CCT/TRiC were pooled, concentrated to 500 µL, and loaded on a gel-filtration (GE Healthcare Superose 6 10/300 GL) column. Fractions containing pure protein were combined and buffer-exchanged on a PD10 column (GE Healthcare) to Tris buffer (pH 7.5) containing 50 mM KCl, 10 mM MgCl$_2$, 2 mM DTT, and 20% glycerol. The protein was then concentrated, divided into aliquots, flash-frozen, and stored at −80 °C until further use.

**Steady-state ATPase measurements**. Initial (steady-state) rates of the ATP hydrolysis were measured as described[16] by monitoring time-resolved changes in fluorescence emission at 475 nm of coumarin-labeled phosphate-binding protein (PBP) upon excitation at 430 nm using an ISS PCI fluorimeter. PBP was expressed by growing *E. coli* BL21 cells, which harbor the pET22b plasmid containing the gene for PBP, until an O.D. of 0.8 was reached and then adding 0.5 mM IPTG. The cells were then grown overnight, harvested by spinning at 11,920 g for 15 mins, and then resuspended in 30 mL of buffer containing 10 mM Tris-HCl (pH 8.0), 1 U DNase1, lysozyme (100 µg/mL), 1:100 protease cocktail inhibitor (Apex Bio, K1007), 2 mM MgCl$_2$ and 2 mM DTT. The resuspended cells were lysed by passing through a French press at 1500 atm. Cell debris was then removed by spinning the lysed cells at 38,720 g for 30 mins and the supernatant was diluted to a final volume of 100 mL with resuspension buffer. The diluted supernatant was then loaded onto a HiTrap Q HP column that had been equilibrated with a resuspension buffer. After washing the column with 50 mL of resuspension buffer, the protein was eluted with a continuous gradient of 0-200 mM NaCl in the same buffer using a total elution volume of 250 mL. Protein-containing fractions were collected and analyzed for purity using SDS-PAGE. Fractions containing PBP were pooled, concentrated, and stored at −80 °C for future use. Labeling of PBP was achieved as before[16]. Nucleotide solutions used in experiments with PBP were treated with phosphate mop (500 µM 7-methylguanosine, 1 U PNPase) as described[16] and the PNPase was then removed by filtering the nucleotide solution on a Centricon YM-50 concentrator. The reactions were carried out at 25 °C in 50 mM Tris–HCl buffer (pH 7.5) containing 10 mM MgCl$_2$, 50 mM KCl, 20% (v/v) glycerol, and 1 mM DTT. The concentration of CCT/TRiC was 20 nM. Changes in fluorescence were converted into changes in phosphate concentration using calibration curves constructed by measuring the fluorescence of labeled PBP at different phosphate concentrations. The data were fitted to the following equation[27] for two allosteric transitions:

$$V_0 = \frac{V_{\max(1)} + V_{\max(2)}\left(\frac{[S]}{K_2}\right)^m}{1 + \left(\frac{K_1}{[S]}\right)^n + \left(\frac{[S]}{K_2}\right)^m} \qquad (1)$$

where [S] is the substrate (ATP) concentration, $V_{\max(1)}$ and $V_{\max(2)}$ are the respective maximal initial rates of ATP hydrolysis by a single ring, and by both rings of CCT, $n$, and $m$ are the Hill coefficients for ATP binding to the first and second rings, respectively, and $K_1$ and $K_2$ are the respective apparent binding constants of ATP for the first and second rings.

α-lactalbumin used in the steady-state and transient kinetic ATPase assays was prepared essentially as described before[19]. In brief, acid-denatured bovine α-lactalbumin was prepared by dissolving 1.75 mg of the protein in 2.5 mL of 0.01 M HCl and

incubating for 30 mins. Calcium ions were then removed by passing the denatured protein through a Sephadex G-25 PD-10 column, which had been pre-equilibrated with 0.01 M HCl. The Ca$^{++}$-depleted protein was then passed through another G-25 PD-10 column, which had been equilibrated with ATPase reaction buffer containing 1 mM DTT. The final concentration of non-folded α-lactalbumin was determined as described[28].

**Stopped-flow experiments**. Reactions were initiated by mixing equal volumes of CCT/TRiC (35 nM oligomer) and PBP (8 µM) with different concentrations of ATP (or various concentrations of α-lactalbumin at fixed final concentrations of 20 or 180 µM ATP) using an Applied Photophysics SX.17MV stopped-flow machine. The reactions were carried out in 50 mM Tris-HCl buffer (pH 7.5) containing 50 mM KCl, 10 mM MgCl$_2$, 1 mM DTT, and 20% glycerol in HPLC (phosphate-free) water at 25 °C. Phosphate release upon ATP hydrolysis was followed by excitation at 430 nm and measuring the fluorescence of the coumarin-labeled PBP at wavelengths higher than 455 nm (using a cutoff filter). A 0.2-cm path length was used, and the entrance monochromator wavelength band pass was set to 9 nm. We averaged 5–10 traces (using a split time base with 1000 data points in the first second and 1000 data points in the remaining time) for each concentration of ATP. The averaged traces were fitted to Eq. S14 (Supplementary Note 1) using Origin. Changes in fluorescence were converted into changes in phosphate concentration using calibration curves constructed by measuring the fluorescence of labeled PBP at different phosphate concentrations. The slope of the calibration curve was found to depend on the precise value of the photomultiplier voltage. Therefore, we determined the dependence of the value of the slope of the calibration curve on the photomultiplier voltage as described[17] so that the data from each experiment could be converted appropriately.

**Dynamic light scattering (DLS) experiments**. The Wyatt DynaPro Nanostar II instrument was employed for DLS measurements. This instrument uses light scattering at 658 nm and a scattering angle of 90°. Protein samples of 20 µL in 50 mM Tris-HCl buffer (pH 7.5) containing 50 mM KCl, 10 mM MgCl$_2$, 1 mM DTT, and 20% glycerol were loaded in a small disposable cuvette for each measurement. The measurements were carried out at 20 °C.

**Autophagy assays**. Cells containing wild-type CCT/TRiC or CCT/TRiC with the double mutation T394P/R510H in subunit CCT2 were transformed with the plasmid pRS315- CUP9-RNQ1-YFP, which directs the expression of RNQ1-YFP under the control of the CUP9 promoter[29]. Single colonies were picked and grown for 24 h in synthetic defined (SD)-leu medium containing 0.5 mM CuSO$_4$. The cells were then spun down and resuspended in SD-leu medium with or without 0.4 µg/mL rapamycin. The cells were then grown for 4 h and imaged using an Olympus IX83 microscope coupled to a Yokogawa CSU-W1 spinning-disc confocal scanner with dual prime-BSI sCMOS cameras 18 (Photometrix). The 16-bit images were acquired with two illumination schemes of 30 ms exposure each: one for brightfield and one for green fluorescence (YFP). For green fluorescence, we excited the sample with a 488 nm laser (Coherent 150 mW) and collected light through a bandpass emission filter (525/50 nm, Chroma ET/m). Imaging was performed with a ×60/1.42 numerical aperture, oil-immersion objective (UPLSAPO60XO, Olympus). Following the acquisition, cells were identified, and segmented and their fluorescent signal was identified using previously reported scripts in ImageJ[30]. Montages of cells were obtained using scripts reported before[31]. The amount of material

in vacuoles was calculated using ImageJ[32] by selecting vacuolar areas and quantifying signal intensity in a random set of cells from the obtained fluorescent images.

**Statistics and reproducibility**. In the case of the stopped-flow experiments, 5–10 traces were averaged for each experimental condition and the data were fitted to Eq. S14. Good fits were obtained as indicated by random deviations about zero of the residuals. In the case of the steady-state ATPase assays, all experiments were carried out at least in duplicate. The error bars represent the standard errors of the slopes obtained from fits to a linear equation.

**Reporting summary**. Further information on research design is available in the Nature Portfolio Reporting Summary linked to this article.

## Data availability

All the data supporting the conclusions of this work are included in the published article and its supplementary information files. Source are available at Figshare: https://figshare.com/s/b859a263cad277b3a7dc.

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

## Acknowledgements

We thank Prof. Douglas Cyr for the kind gift of the pRS315- CUP9-RNQ1-YFP plasmid and Prof. Zvulun Elazar for helpful advice. This work was supported by the Minerva Foundation with funding from the Federal German Ministry for Education and Research. A.I.A. received funding from grant # 075-15-2021-1071 for the development of genomic editing technologies for innovation in industrial biotechnology from the Ministry of Science and Higher Education of the Russian Federation. A.H. is grateful for support from the Helen & Milton A. Kimmelman Center for Biomolecular Structure and Assembly and the Ilse Katz Institute for Material Sciences and Magnetic Resonance Research. A.H. is an incumbent of the Carl and Dorothy Bennett Professorial Chair in Biochemistry.

## Author contributions

A.H. conceived the project and wrote the first draft. M.R. and R.C.F. conducted all the experiments. A.I.A. was involved in the imaging experiments and analysis of the yeast cells. M.R., R.C.F., A.I.A. and A.H. revised the manuscript. All authors read and approved the final version of the manuscript.

## Competing interests

The authors declare no competing interests.
