## [Peer Review File · Communications Biology]

Reviewers' comments:

Reviewer #1 (Remarks to the Author):

Horovitz and colleagues investigated the kinetic behavior of double-mutated CCT2 in CCT/TRiC complex from yeast, the heterozygous double mutation in CCT/TRiC was previously reported to link to an inherited retinal dystrophy affecting 20% of childhood blindness cases. Based on the steady-state and transient kinetic analysis on the yeast CCT, they propose that the double mutation stabilizes the closed state of CCT/TRiC and therefore impair its function in autophagy. Given the importance of the disease implication of CCT mutations, the work may represent a useful clue for further understanding of CCT structure and function and for potential disease intervention. The authors may address the following concerns in a revision before any recommendation on publication can be made.

(1) It would be nice to show in a main figure or figure panel where the two mutations are located in the structure of CCT/TRiC complex, given there are already CCT/TRiC structure determined by cryo-EM available.

(2) The authors may want to provide certain discussion on the inter-residue interactions around the two mutated residues to support their speculation on how the two mutations stabilize the close conformation. Such discussion or argument should be accompanied by one or few figure panels, in which the interfaces between the closed and open states should be illustrated.

(3) Because the disease implication of LCA was reported for human or mammalian species while the current analysis was done on the yeast CCT, the authors should show, preferably in a main figure panel, the amino acid sequence alignment between yeast and human, as well as other closely related species commonly used in similar studies (such as bovine and mice, etc.). This is expected to show how conserve the sequence at the location of the mutations among different species and therefore how much we can extrapolate from the current studies on the yeast proteins.

(4) Based on the above additional analysis, the authors should briefly discuss the limitations and caveats in speculating the underlying allosteric mechanism due to lack of direct structural determination, in extrapolating the yeast results to the human diseases. The authors should avoid confusing hypothesis-based speculation with evidence-based conclusion.

(5) The title of this paper is misleading and not supported by the data at all, because the work was done on the yeast proteins and the disease of Leber congenital amaurosis is related to the human or mammalian ortholog of CCT. There are no human or mammalian CCT analyzed in the current study and therefore no conclusion should be made to the human CCT, and the inference of yeast CCT results to the case of mammalian CCT is purely hypothetical and lack of any direct evidence.

Reviewer #2 (Remarks to the Author):

The manuscript titled as "The double mutation T400P/R516H in subunit CCT2 of TRiC/CCT reduces the off-rate of ADP thereby causing Leber congenital amaurosis" by Roy et al describes the diminished ADP off rate of LCA associated mutant (T400P and R516H) of CCT2 subunit of TRiC/CCT complex using its yeast orthologue. The study exclusively shows the decreased ADP off rate by the double mutant CCT2 in comparison to the wild type or the single mutants by performing ATPase assays. The claims of the authors to explain the molecular mechanism of LCA due to these two mutations in CCT2 is inadequately supported by the experimental evidences and majorly based on speculations. Thus, the manuscript is not suitable for its publication in Communication Biology. The comments are mentioned below

1. The authors claim that the double mutant CCT2 is inefficiently dissociated from the TRiC/CCT complex due to stabilization of the closed complex due to its slower ADP off rate. This conclusion is made from binding energy calculation and not supported by any experiments. The slower dissociation of the double mutant of CCT2 from the TRiC/CCT complex should be shown experimentally.

2. The authors also assume that the double mutant of CCT2 due to its less dissociation from TRiC/CCT complex will be less efficient in autophagic degradation of the aggregates. This point is based on assumption which should be supported by experiments in the yeast model.
3. The title of the paper is misleading as there is no experimental evidence apart from the slower off rate of ADP.

Reviewer #3 (Remarks to the Author):

This is a very well conducted and important study on the biochemistry of the CCT ATP cycle which is perturbed in the CCT2 double mutation responsible for CCT2-Related Leber Congenital Amaurosis.

1. I think the title should be inverted to something like - The double mutation T400P/R516H in subunit CCT2 of TRiC/CCT which causes Leber congenital amaurosis reduces the off-rate of ADP.
2. This paper should certainly be referenced and discussed: Mutation in the Zebrafish cct2 Gene Leads to Abnormalities of Cell Cycle and Cell Death in the Retina: A Model of CCT2-Related Leber Congenital Amaurosis. *Investigative Ophthalmology & Visual Science*, 01 Feb 2018, 59(2):995-1004 DOI: 10.1167/iovs.17-22919. The cct2-L394H-7del mutant deletes the helix including T400. I note that this study does not include any transient kinetic analysis of ATP hydrolysis or turnover.
3. alpha-lactalbumin assay. I couldn't establish the ratio of CCT to substrate in this measurement at fixed [ATP] of 300 micromolar. Is there any evidence that the CCT is dissociated or destabilised by this substrate binding?
4. Are there any effects on growth of yeast carrying these mutations?
5. A diagram to illustrate the heterozygous/homozygous combinations in the human condition and the yeast a zebrafish mutants would aid the non-expert I think.

Amnon Horovitz, Professor

Incumbent of the Carl and Dorothy Bennett Chair
in Biochemistry
Dept. of Chemical and Structural Biology
Weizmann Institute of Science
Rehovot 76100, Israel

July 19, 2023

Dear Editor and Reviewers,

Thanks very much for the reviews of our paper and for the opportunity to revise it. We found many of the comments of the Reviewers very useful and have addressed them as detailed below. The revised paper contains six new figures (one in the main text and five in the supplementary material) of which three show new experimental data. A co-author who assisted in the autophagy assays was added. The Reviewers' comments and our responses follow.

Reviewer #1 (Remarks to the Author):

(1) It would be nice to show in a main figure or figure panel where the two mutations are located in the structure of CCT/TRiC complex, given there are already CCT/TRiC structure determined by cryo-EM available.

Response: A main figure (Fig. 1 in the revised paper) was added that includes a panel showing the structure with the mutation sites.

(2) The authors may want to provide certain discussion on the inter-residue interactions around the two mutated residues to support their speculation on how the two mutations stabilize the close conformation. Such discussion or argument should be accompanied by one or few figure panels, in which the interfaces between the closed and open states should be illustrated.

Response: The new Fig. 1 shows how the mutated residues may interact with the nucleotide. We have also added a new supplementary Fig. 4 that shows contact maps of CCT2 with its neighboring subunits. This figure shows very clearly that CCT2 has many more inter-subunit contacts in the closed state. Hence, a more long-lived closed state will impede the exit of CCT2 from the complex.

(3) Because the disease implication of LCA was reported for human or mammalian species while the current analysis was done on the yeast CCT, the authors should show, preferably in a main figure panel, the amino acid sequence alignment between yeast and human, as well as other closely related species commonly used in similar studies (such as bovine and mice, etc.). This is expected to show how conserve the sequence at the location of the mutations among different species and therefore how much we can extrapolate from the current studies on the yeast proteins.

Response: We thank the reviewer for this suggestion. The conservation of helices 14 and 18, which contain the mutation sites, in yeast and human CCT2 is shown in panel b of the new Fig. 1. The legend contains a statement about the overall conservation of CCT2. In panel c, it

shown that the structures of human and yeast CCT2 are very similar with a root mean square deviation of about 1.2 Å.

(4) Based on the above additional analysis, the authors should briefly discuss the limitations and caveats in speculating the underlying allosteric mechanism due to lack of direct structural determination, in extrapolating the yeast results to the human diseases. The authors should avoid confusing hypothesis-based speculation with evidence-based conclusion.

Response: We point out more clearly in the revision that the allosteric mechanism involves inter-ring communication and that this is also supported by the fact that the CCT2 subunits in the two rings interact directly. We also point out that the molecular mechanism underlying this inter-ring communication is not known.

(5) The title of this paper is misleading and not supported by the data at all, because the work was done on the yeast proteins and the disease of Leber congenital amaurosis is related to the human or mammalian ortholog of CCT. There are no human or mammalian CCT analyzed in the current study and therefore no conclusion should be made to the human CCT, and the inference of yeast CCT results to the case of mammalian CCT is purely hypothetical and lack of any direct evidence.

Response: We agree with the reviewer and adopted a modified version of the title suggested by Reviewer 3. The new title is: 'The double mutation T394P/R510H in yeast CCT2 which causes Leber congenital amaurosis in humans reduces the ADP off-rate'.

Reviewer #2 (Remarks to the Author):

The manuscript titled as "The double mutation T400P/R516H in subunit CCT2 of TRiC/CCT reduces the off-rate of ADP thereby causing Leber congenital amaurosis" by Roy et al describes the diminished ADP off rate of LCA associated mutant (T400P and R516H) of CCT2 subunit of TRiC/CCT complex using its yeast orthologue. The study exclusively shows the decreased ADP off rate by the double mutant CCT2 in comparison to the wild type or the single mutants by performing ATPase assays. The claims of the authors to explain the molecular mechanism of LCA due to these two mutations in CCT2 is inadequately supported by the experimental evidences and majorly based on speculations. Thus, the manuscript is not suitable for its publication in Communication Biology. The comments are mentioned below

1. The authors claim that the double mutant CCT2 is inefficiently dissociated from the TRiC/CCT complex due to stabilization of the closed complex due to its slower ADP off rate. This conclusion is made from binding energy calculation and not supported by any experiments. The slower dissociation of the double mutant of CCT2 from the TRiC/CCT complex should be shown experimentally.

Response: We were not sufficiently explicit before. We note in the revised version that our experimental data reveal a two-fold slower ADP off-rate from which the amount of free CCT2 can be estimated to be reduced by a factor of two.

2. The authors also assume that the double mutant of CCT2 due to its less dissociation from TRiC/CCT complex will be less efficient in autophagic degradation of the aggregates. This point

is based on assumption which should be supported by experiments in the yeast model.

Response: Following the Reviewer's suggestion, we tested whether autophagy is impaired in the yeast using RNQ1-YFP as a model protein aggregate. We found that addition of rapamycin leads to an increase in YFP found in the vacuoles of cells with wild-type CCT/TRiC but not with the double mutant. The results are shown in the new Supplementary Fig. 1.

3. The title of the paper is misleading as there is no experimental evidence apart from the slower off rate of ADP.

Response: We agree with the reviewer and adopted a modified version of the title suggested by Reviewer 3. The new title is: 'The double mutation T394P/R510H in subunit CCT2 of yeast TRiC/CCT which causes Leber congenital amaurosis in humans reduces the off-rate of ADP'

Reviewer #3 (Remarks to the Author):

1. I think the title should be inverted to something like - The double mutation T400P/R516H in subunit CCT2 of TRiC/CCT which causes Leber congenital amaurosis reduces the off-rate of ADP.

Response: We thank the reviewer for his suggestion which we accepted but modified to emphasize that the work was carried out in yeast. The new title is: 'The double mutation T394P/R510H in subunit CCT2 of yeast TRiC/CCT which causes Leber congenital amaurosis in humans reduces the off-rate of ADP'.

2. This paper should certainly be referenced and discussed: Mutation in the Zebrafish cct2 Gene Leads to Abnormalities of Cell Cycle and Cell Death in the Retina: A Model of CCT2-Related Leber Congenital Amaurosis. Investigative Ophthalmology & Visual Science, 01 Feb 2018, 59(2):995-1004 DOI: 10.1167/iovs.17-22919. The cct2-L394H-7del mutant deletes the helix including T400. I note that this study does not include any transient kinetic analysis of ATP hydrolysis or turnover.

Response: We've added a citation and short discussion of this paper.

3. alpha-lactalbumin assay. I couldn't establish the ratio of CCT to substrate in this measurement at fixed [ATP] of 300 micromolar. Is there any evidence that the CCT is dissociated or destabilised by this substrate binding?

Response: The X-axis of Fig. 4 indicates the α -lactalbumin concentration and the concentration of CCT, which is fixed, is stated in the legend to this figure. In order to address the concern regarding possible destabilization of CCT upon α -lactalbumin binding, we carried out dynamic light scattering experiments. The results which are shown in the new Supplementary Fig. 3 indicate that there is no destabilization.

4. Are there any effects on growth of yeast carrying these mutations?

Response: The effects on cell growth are relatively small as shown in the new Supplementary Fig. 5.

5. A diagram to illustrate the heterozygous/homozygous combinations in the human condition and the yeast a zebrafish mutants would aid the non-expert I think.

Response: Thanks for this suggestion. A scheme showing the combinations of mutations found in patients and in our study is provided in the new Supplementary Fig. 2.

In summary, we believe that the review process has greatly improved our paper. Thanks for your attention.

Best wishes,

Prof. Amnon Horovitz

REVIEWERS' COMMENTS:

Reviewer #1 (Remarks to the Author):

The authors have address all my questions except for one minor issue. The revised title might be still misleading. It reads like "yeast CCT2 double mutation causes Leber congenital amaurosis in human", which sounds like the LCA is mediated through "yeast infection". But what I understand is that the human ortholog of the yeast mutation can cause LCA disease. If I am correct, the author should revise the title again to avoid such a confusion. Other than this, I recommend its publication after minor revision.

Reviewer #3 (Remarks to the Author):

The new experiments and figures have really improved this study overall and the manuscript is much easier to navigate. Great work.